# Diversity in Education Study (DivES): Investigating a neurodiversity module in higher education

Amy J. Schwichtenberg*, Katherine Mirah, Amy Janis, Moon West, Annabelle L. Atkin

Department of Human Development and Family Science, Purdue University, West Lafayette, Indiana, United States of America

* ajschwichtenberg@purdue.edu

## Abstract

Roughly 20% of adults identify as neurodivergent – an umbrella term used to describe cognitively atypical individuals. Neurodivergent identities manifest in several forms including autism, attention dysregulation hyperactivity disorder/attention deficit hyperactivity disorder, dyslexia, and other cognitively atypical trajectories. Despite a societal increase in acknowledgment and awareness of neurodiversity, there remains little to no discussion or critical understanding of neurodivergent individuals, especially within the realm of higher education (HE). The aim of this study, as part of the Diversity in Education Study (DivES), is to explore the impact of a neurodiversity-focused module within the college classroom. This study included 153 students (predominantly female, heterosexual, third-year students, with white/European racial heritage); wherein, a neurodiversity-focused module was taught within the context of an undergraduate course on diversity. Pre (start of the term) and post (end of the term) surveys were compared to evaluate the impact of the module on student understanding of the term neurodiversity, self-endorsements of a neurodivergent identity, and critical consciousness of ableism. In sum, the neurodiversity-focused module increased neurodiversity awareness and self-endorsements but did not positively shift critical consciousness of ableism scores. Replication in larger and more diverse samples is needed before pedagogy recommendations may be solidified but this study provides preliminary support for the incorporation of a neurodiversity module within diversity-focused HE courses.

## Introduction

Rooted in the social model of disability, the neurodiversity paradigm was created to challenge the pathologizing nature of medical model of disability [1–3]. Neurodiversity, conceptually, highlights the wide variety of neurological development as an important facet of the human condition rather than pathologizing so called "atypical"

**Data availability statement:** All data associated with this paper are publicly available at https://osf.io/p98dc/ (doi:10.17605/OSF.IO/P98DC).

**Funding:** The author(s) received no specific funding for this work.

**Competing interests:** The authors have declared that no competing interests exist.

ways of thinking [1,3,4]. In its infancy, neurodiversity was primarily used to describe individuals with autism; however, it is now commonly understood as an umbrella term that includes autism, dyslexia, attention dysregulation hyperactivity disorder/attention deficit hyperactivity disorder (ADHD) [5] dyscalculia, dyspraxia, and other diagnoses [6]. Recently the term has also been applied to individuals with chronic health conditions that can impact cognition (e.g., chronic pain conditions).

In the context of higher education (HE), there has been an intentional push to enroll and accommodate more diverse populations. Yet, despite this effort, neurodiversity is not often included in discussions surrounding diversity. Previous research has indicated the neurodiversity paradigm needs to be a topic of discussion in higher education to make these environments more accepting and supportive of neurodivergent individuals [7–10]. This can take many forms, such as including neurodiversity in diversity, equity, and inclusion (DEI) initiatives seen throughout HE [11]. Neurodivergent individuals often report negative experiences in HE [6,9,12]. Many report feeling or being treated merely as a label rather than a person [7]. Additionally, the challenge of pursuing HE can be amplified for those who identify as neurodivergent as they are often quickly subjected to ableism from peers and HE policies [7,10].

Several studies have investigated how neurodivergent individuals interact with HE institutions and consequently how they are treated within the contexts of these institutions [9,13]. A common recommendation is to educate individuals about ableism in HE and the lived experiences of neurodivergent individuals [11,14]. However, there is little research regarding the implications of teaching college-level students about the neurodiversity paradigm and how this may have an impact on students' perceptions, attitudes, and beliefs surrounding neurodiversity. The current study is an attempt to further the discussion of diversity, specifically focusing on the effectiveness of teaching students about the neurodiversity paradigm within a university setting. The present study builds on two literature areas – ableism and lived experiences of neurodivergent individuals in HE.

### Ableism in higher education

Ableism is a pervasive societal issue that negatively effects neurodivergent populations [15]. As defined by disability scholar Fiona Kumari Campbell ableism is "a network of beliefs, processes and practices that produce a particular kind of self and body (the corporeal standard) that is projected as the perfect, species-typical and therefore essential and fully human [16]." Ableism includes discriminatory actions or policies that systematically favor those perceived as able-bodied or able-minded. It can include acts reflecting fear, contempt, or pity. Ableism and ableist ideologies are found in many aspects of society, one being higher education [17,18]. The ability to recognize these systems of inequity (critical consciousness of ableism) is a key area of growth highlighted by many scholars and areas of disability studies.

Although ableism and its effects are felt throughout the education system; it is often not recognized as a significant issue [10]. Ableism deeply impacts individuals with disabilities, including those who identify as neurodivergent. Those with disabilities often describe negative experiences and a lack of support within schools which

can in turn can lead to poor academic performance and an increased likelihood of withdrawing from HE altogether [10]. Due to negative experiences early in the education system, neurodivergent individuals can be discouraged from disclosure [6] and often go on to mask their adult symptoms in the workplace and health care settings [19–21].

Ableism is felt by a large portion of neurodivergent individuals in HE, including faculty in HE that identify as neurodivergent [22]. There is a normalized work culture that emphasizes productivity and demanding work loads [7]. This culture makes it difficult for individuals to feel comfortable sharing their experiences. Several faculty members within HE who identify as neurodivergent reflect that disclosing their neurodivergent identity often comes with fear of diminished academic reputation and reductions in perceived competence [7]. The sheer number of individuals that choose not to disclose their disability, due to fear of stigmatization, demonstrates that society, including the HE system, is infused with ableism [7,10,17].

## Lived experiences in higher education

Entering HE can be challenging for many individuals resulting in a variety of experiences, reactions, and feelings. When neurodivergent individuals enter HE, many of their reported experiences, reactions, and feelings are often described as negative [6,12,13].

Clouder et al. (2020) conducted a systematic review of 48 studies investigating neurodivergent individuals in HE and reported a recurring theme of anxiety [6]. Neurodivergent students in HE often endorse feeling anxious both in and out of the classroom as they are unsure of how their peers and professors will respond to them – especially if they disclose their disability. Many may have had negative experiences when interacting with the education system prior to entering HE, which leads them to assume future experiences will have a similar outcome. Additionally, traditionally demanding learning environments in HE often include detailed course elements and strict rules that can be difficult for neurodivergent individuals to navigate. The current norm in HE courses, and the method they are delivered, may cause these students to feel overwhelmed and unsupported. Individuals who identify as neurodivergent are also often subjected to microaggressions and discrimination regarding their disability and have an increased likelihood of being cyberbullied, which can lead to further negative experiences in HE [23,24]. For example, in a study of ADHD medical students, experiences of microaggressions and discrimination lead to feelings of isolation from peers, educators, and the educational institution [25]. Similarly, dyslexic medical students reported feeling hopeless and helpless due to experiences of self-consciousness, shame, and isolation [26].

Often reinforced by systematic ableism, internalized ableism also has harmful effects on the lives of disabled individuals. Griffin and Pollak (2009) identified that individuals who described having a disability as a deficit and utilized negative words to describe their disability had worse academic outcomes than individuals who described their disability as a difference that holds both positive and negative aspects [12]. Reducing systemic ableism is a clear step towards supporting neurodivergent individuals within HE contexts. Neurodivergent students may feel more supported in HE if neurotypical individuals were taught about neurodiversity. This increased knowledge can support neurotypical individuals in their ability to identify systematic ableism in HE as well as its impact on their neurodivergent peers [7,11,14]. This can further improve awareness and support to neurodiverse students. Therefore, an essential step forward is to educate others on neurodiversity and ableism.

## Current study

Within an existing undergraduate course on diversity, a weeklong module on neurodiversity was added and pre- (start of the term) and post- (end of the term) surveys were compared across two terms. The current study addressed the research questions of (1) will student knowledge in neurodiversity and self-disclosure of a neurodivergent identity increase over the course of the term and (2) will the neurodiversity module shift reports of critical consciousness of ableism? We hypothesized increases (comparing pre- to post-survey responses) in student knowledge in neurodiversity, student self-endorsement of neurodivergent identities, and reports of critical consciousness in ableism.

## Positionality statement

Positionality can introduce bias. Therefore, we've provided a summary of our experiences and/or affiliations that may be relevant to the current study. Five authors worked on the current study. Two of the authors are currently faculty who have taught courses relating to atypical child development and diversity. One directs an autism research center and has extensive research experience in neurodevelopmental and neurodegenerative disorders. The other professor has research interests that include racial attitudes, racial-ethnic socialization/identity, discrimination, and critical consciousness. The other three authors are students with a range of research interest (e.g., sleep, neurodevelopment disorders, equitable school practices). Three authors on this team identify as neurodivergent. Across all authors, there is a mutual interest in increasing neurodiversity knowledge and inclusivity in HE.

## Methods

### Participants

The participants of the study $(N = 153)$ were enrolled in a diversity course at Purdue University. The study included a six-part survey completed at the start and end of the term. The diversity course was taught by two professors, Professor X $(n = 103)$ and Professor Y $(n = 50)$ and was offered in Fall 2021 and Spring 2022. The participants ranged from first-year students to fifth-year students, with most of them being second-year students, $M(SD) = 2.57(1.10)$ years. Most students identified as female $(n = 134, 88\%)$ as well as straight/heterosexual $(n = 121, 79\%)$. In terms of race, most students endorsed White/European American only $(n = 122, 80\%)$ and the rest endorsed a racial/ethnic minority or multiracial heritage $(n = 31, 20\%)$. The age of participants varied, ranging from seventeen to twenty-seven years with a majority reporting roughly twenty. For demographic information see Tables 1 and 2.

### Procedure

This study is part of the Diversity in Education Study (DivES) project at Purdue University (approved by the Purdue University Institutional Review Board - IRB-2021–1054). This study was exempted from full IRB review under category two, which allows for the waiver of informed consent procedures when data is collected via an educational survey. However,

**Table 1. Sample demographic information.**

| Variable | Response n | n (%) | Min-Max | M(SD) |
|---|---|---|---|---|
| Gender Identity | 153 | | | |
| Female | | 134 (87.6) | | |
| Male | | 16 (10.5) | | |
| Non-binary | | 3 (2) | | |
| Sexual Orientation | 153 | | | |
| Straight/heterosexual | | 121 (79.1) | | |
| Bisexual | | 15 (9.8) | | |
| Gay/lesbian | | 8 (5.9) | | |
| Pansexual | | 4 (2.6) | | |
| Asexual | | 2 (1.3) | | |
| Unsure | | 2 (1.3) | | |
| Age (years) | 146 | | 17-27 | 19.96 (1.51) |
| Academic Year [a] | 153 | | 1-5 | 2.57 (1.10) |

Note. [a]For academic year 1, 2, 3, and 4/5 correspond with Freshman, Sophomore, Junior, and Senior in college, respectively.

to ensure transparency and respect for participant autonomy, the first question within the survey asks for explicit student informed written consent. This consent form outlined the study's purpose, length, potential risks and benefits, and confidentiality. Opportunities for consent-related questions were provided in-person during class. To maintain the integrity of the study, instructors were blind to student consent status (i.e., they knew all students completed the survey for class but were not aware of who consented for their data to be used within this study). The study utilized a pre-post-design using Qualtrics© (Qualtrics, Provo, UT). Data were stored, de-identified, and shared by the Office of Undergraduate Research. All data were used for university programming but only those who consented ($n = 153$) are included in this research report. The six-part survey included questions assessing social issue attitudes (e.g., racism, ableism), racial experiences (e.g., racial discrimination), engagement in social action, identity, and mental health. A portion of the survey contained a series of questions on neurodiversity and ableism. The ableism questions utilized fourteen statements adapted from the Contemporary Critical Consciousness Measure II (CCCMII) [30]. For the larger DivES study recruitment began August 15, 2021, and is still ongoing. The data selected for this report includes all consented students between the recruitment period of 15/08/2021 to 15/05/2022. Functionally, this was Fall, 2021 and Spring, 2022 terms at Purdue University (the first year of the DivES study).

## Neurodiversity module

One week of the 16-week course was dedicated to students engaging with material related to neurodiversity. Before attending the lectures for that week, students were required to read Chapter 6, Understanding Privilege Through Ableism, from the textbook, *Is Everyone Really Equal: An Introduction to Key Concepts in Social Justice Education*, Second Edition written by Ozlem Sensoy and Robin DiAngelo (2012) as well as the article *Neurodiversity: An insider's perspective* written by Jacquiline den Houting (2019). Additionally, students were required to listen to Radiolab's podcast *Juicervose* produced by Kelsey Padgett. Students were asked to complete a comprehension quiz on the assigned readings and podcast prior to lecture. They were given the choice of reading *Sibling Relationships Over the Life Course – Growing up with a Disability* by Hila Avieli, Tova Band-Winterstein, and Tal Araten Bergman (2019).

One class period was dedicated to a presentation on neurodiversity. Information was presented on the Disability Rights Movement, Neurodiversity Movement and different models and ideas related to disabilities and neurodiversity. Another class period, students were asked to either (a) attend a neurodiversity panel and listen to neurodivergent individuals' experiences or (b) engage in an in-person session with a neurodivergent peer who attends an alternative life skills university. The in-person session had a social focus and students were asked to engage with neurodivergent peers. The hope was that by learning about neurodiversity, individuals would reaffirm or change their thoughts and opinions regarding neurodiversity, and they would then display that reaffirmation or change in their post-survey responses.

**Table 2. Sample race and ethnicity information.**

| Race | nᵃ | Endorsed multiple categories |
|---|---|---|
| African American | 10 | 6 |
| Asian American/Asian | 12 | 2 |
| Native Hawaiian or Pacific Islander | 1 | 1 |
| White/European American | 133 | 12 |
| Hispanic/Latinx | 12 | 8 |
| Native American/American Indian | 1 | 1 |

Note. ᵃ Individuals could endorse more than one category.

## Critical consciousness of ableism

The statements utilized to assess individuals' attitudes towards ableism were adapted from the Contemporary Critical Consciousness Measure II (CCCMII) [27]. Students were asked on a scale of 1–7, with 1 being "Strongly Disagree" and 7 being "Strongly Agree," whether they agreed or disagreed with the ableism statements and to what extent. These responses were then scored and summed to gauge an individual's critical consciousness, or the extent to which they are aware of the ableism that is infused in various aspects of our society [27]. Sample statements include, "Discrimination against people with physical disabilities is a major problem in U.S. society" and "People with disabilities are generally treated more poorly in U.S. society than those without disabilities" [27]. Consistent with previous studies that utilize the CCCMII [28,29], high critical consciousness is seen when an individual obtains a higher score. The max high score is 98 (high critical consciousness) and the lowest score possible is 14 (low critical consciousness). This measure had high reliability - Conbach's α and McDonald's Ω = .91 in the pre survey and both were .93 in the post survey. Distribution checks (histogram, skewness, and kurtosis inspections) confirmed a normal distribution of scores (pre skewness = .43, kurtosis = .49; post skewness = .34, kurtosis = −.55). One outlier was discovered (student provided the highest score for all items) in the pre survey but this same student provided more variable/typical data in the post survey. Therefore, within the pre survey mean imputation was utilized instead of the outlier score.

## Neurodivergent identity

The participants were asked if they consider themselves neurodivergent, with response options including "yes," "no," "I don't know what neurodiverse means," and "prefer not to answer." If they indicated yes, they were also asked "How are you neurodiverse?" The responses were read and coded into one of following categories, (1) ADHD, ADD, and attention problems; (2) autism; (3) sensory sensitivities; (4) depression and anxiety; (5) other psychological disorders, including obsessive-compulsive disorder (OCD), borderline personality disorder, bipolar disorder, emetophobia and other general mental health issues; (6) specific learning disabilities, such as dyslexia; and (7) individuals who overall think differently than others. Some responses clearly indicated a category, such as "I have ADHD," whereas other responses were coded into a category due to key words or ideas. For example, "I think differently than others" and "I perceive things differently than others" would both be coded into the "think differently than others" category. For those who endorsed being neurodivergent, the specific ways they identified are summarized in Fig 1.

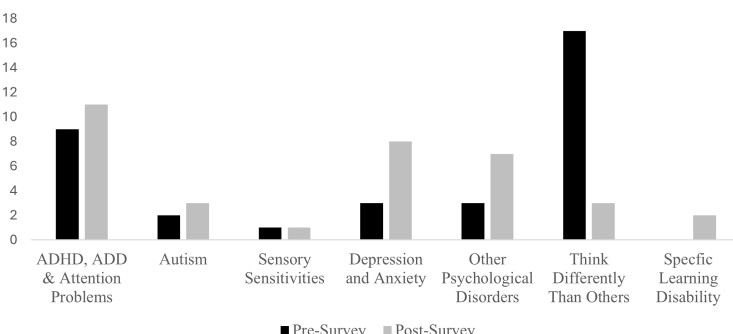

**Fig 1. Individual neurodiversity endorsements in the pre- and post- surveys.** Note. Other Psychological Disorders for the Pre-Survey includes obsessive-compulsive disorder (OCD) and other general mental health concerns. For the Post-Survey Other Psychological Disorders include OCD, bipolar disorder, emetophobia, and other general mental health concerns. Specific Learning Disability includes dyslexia and other learning disabilities.

## Demographic information

Sample demographic data were pulled from questions on self-reported race, gender, and sexual orientation (summarized in Tables 1 and 2). Each variable was dichotomized based on the most commonly endorsed category (e.g., heterosexual vs. all other sexual orientation identities). In the example of sexual orientation, grouping these smaller categories together allowed for potential differences related to minoritized sexual orientation status to be considered, without overfitting the model or drawing inferences from underpowered (small $n$) multi-group comparisons.

## Analytic plan

Building on reviewer feedback, we assessed missingness patterns. The dataset used for analyses had a total missingness rate of < 2% with fewer than two missing data patterns. Given this, we could not test for missing completely at random (MCAR) but given the low percentage of missing data we were comfortable assuming missingness would not have a robust impact on our pattern of findings. To assess research questions 1, 2, and 3 frequency counts and chi square ($X^2$) analyses were employed. To assess research question 4, paired sample t-scores were utilized.

## Results

### Research question 1: Did the endorsement of a neurodivergent identity vary by gender, sexual orientation, race, or professor?

This research question was posed to confirm if covariates should be included in subsequent neurodivergent identity analyses. Neurodivergent identity self-endorsements did not systematically vary by gender in the pre-survey, $X^2$ (1, N = 150) = .57, $p$ = .45, φ = −.06 or the post-survey, $X^2$ (1, N = 150) = .06, $p$ = .81, φ = −.02. Similarly, endorsement of neurodivergent identities also did not systematically vary by sexual orientation in the pre-survey, $X^2$ (1, N = 153) = 1.45, $p$ = .23, φ = .10 or the post-survey, $X^2$ (1, N = 153) = 1.97, $p$ = .16, φ = .11. Endorsements did not notably differ by race in the pre-survey, $X^2$ (1, N = 153) = 3.62, $p$ = .06, φ = .15 or the post-survey, $X^2$ (1, N = 153) = .35, $p$ = .56, φ = −.05. Endorsements were also comparable across the two professors: pre survey $X^2$ (1, N = 153) = .43, $p$ = .51, φ = .05 and post survey $X^2$ (1, N = 153) = .32, $p$ = .57, φ = −.05.

### Research question 2: Did the neurodiversity module increase neurodiversity knowledge?

In the pre-survey, 70 (46%) of persons selected "I don't know what neurodiverse means," followed by only 5 (3%) in the post-survey. This was a significant decrease, $X^2$ (1, N = 153) = 36.77, $p$ < .001, φ = .49.

### Research question 3: Did the neurodiversity module increase the likelihood of self-disclosing a neurodivergent identity?

In the pre-survey, 19 (12%) students endorsed a neurodivergent identity, followed by 42 (28%) in the post survey. In sum, neurodivergent identity endorsements were significantly higher in the post-survey, $X^2$ (1, N = 153) = 17.85, $p$ < .001, φ = .34.

As illustrated in Fig 1, the types of neurodivergent identities also shifted from pre- to post. Most of the students (14 of 18) who were coded "Think Differently than Others" in the pre-survey, later selected "no" to a neurodivergent status and no longer provided a description how they were neurodivergent in the post survey. The other notable shifts from pre to post were an increase in the two categories of (1) Depression and Anxiety, and (3) Other Psychological Disorders. For those who endorsed "yes" to the neurodivergent question in the post-survey, 16 consistently endorsed "yes" from pre-post, 5 switched from "prefer not to answer" to yes, 14 switched from "I don't know what neurodiverse means" to "yes", and 7 switched form "no" to "yes".

### Research question 4: Did the neurodiversity module shift reports of critical consciousness of ableism?

In the pre-survey, 153 students completed the adapted Contemporary Critical Consciousness Measure II (CCMII) ableism scale, $M(SD)$ = 49.45(8.09), and 151 completed it in the post-survey, $M(SD)$ = 47.16(7.37), across all class sections.

For those who completed the CCCMII ableism critical consciousness scale in both the pre- and post-survey ($n = 151$), a paired-sample t-test was used to assess change from the start to the end of the term. The average CCCMII ableism scale score significantly decreased from the pre- to the post-survey, $t(150) = 3.79$, $p < .001$, Cohen's $d = .31$.

### Instructor differences

The ableism scale findings were unexpected and prompted further (post-hoc) evaluation of scores by instructor. For Professor Y, the mean change in scores was −.73 (i.e., stable from pre to post, $t(48) = .73$, $p = .47$, Cohen's $d = .10$) and for Professor X, mean scores change was −3.01 (i.e., a notable change, $t(94) = 4.11$, $p < .001$, Cohen's $d = .41$). In the case of the CCMII ableism scales, differences by instructor emerged. Differences by instructor were also investigated for the other research questions (see Table 3), but it did not influence the other results.

### Discussion

Within this study, a week-long module increased neurodiversity awareness and comfort in self-disclosing neurodivergent status. However, it was inadequate at increasing one's critical consciousness of ableism (or the ability to recognize systems of inequality that favor abled persons). Although connected, understanding a definition of neurodiversity is not the same as recognized systems of oppression. The neurodiversity module utilized in this study focused on understanding neurodiversity, its development/history, and its real-world applications today. The training aimed to de-pathologize neurodivergent identities and gave students interpersonal experiences with neurodivergent peers. One of the readings included in the neurodiversity module focused on ableism but with no in-person content. This likely contributed to the divergent findings across the outcomes of interest.

When considering the pattern of findings within this study, the larger 'parent' project should be considered. The larger project, Diversity in Education Study (DivES), included educational modules on several systems of oppression, discrimination, and prejudice with respect to race, ethnicity, sexual orientation, and several additional identities. Although systems of ableism were not explicitly taught it seemed reasonable, at the time of study design, that the consciousness of ableism could increase with the consciousness of other systems of oppression, discrimination, and prejudice. Findings from this study support more direct educational approaches, as the more indirect approach employed in DivES may not be sufficient to recognize patterns of ablism in HE.

The rates of neurodivergent identities within this sample may initially appear high (12–28%). At Purdue University (PU), roughly 6–7% of the student body are registered with the Disability Resource Center (DRC). This is a required step in the process of receiving accommodations at PU. However, in Purdue's Student Experience Survey roughly 36% of the incoming student cohort endorsed a category that could be considered a neurodivergent identity (e.g., autism, ADHD, bipolar disorder, dyslexia, dyscalculia). Estimates from the National Center for Educational Statistics [30] document roughly 20% of HE students endorsed similar categories but only 12–15% reported this to their home institution [31]. Therefore, upon reflection of larger patterns, the neurodivergent identity endorsements within this study are not outside of expectations. However, they do draw attention to the large portion of students in HE classrooms who could be served by neurodiversity-affirming practices.

**Table 3. Breakdown of endorsement of a neurodivergent identify by gender, sexual orientation, race, and professor.**

| Variable | Female | | Heterosexual | | White/European | | Professor | |
|---|---|---|---|---|---|---|---|---|
| n (%) | Pre | Post | Pre | Post | Pre | Post | Pre | Post |
| **Neurodiverse** | 17 (11%) | 37 (25%) | 13 (9%) | 30 (20%) | 12 (8%) | 35 (23%) | 14 (9%) | 27 (18%) |
| **Neurotypical** | 116 (78%) | 96 (64%) | 107 (70%) | 90 (75%) | 109 (71%) | 87 (57%) | 88 (60%) | 76 (50%) |

Note. Reflects the most common group in terms of gender, sexual orientation, race, and professor. Each variable is coded in a binary manner.

## Pedagogical reflections

Replication of this study with larger and more diverse samples is necessary to validate these preliminary findings and refine the neurodiversity module. For example, building on the recommendations of others, elements of compassion training could be incorporated to aid in highlighting systems of oppression [32]. Additionally, a social–relational model of disability could be adopted throughout the course [33]. Due to the neurodiversity unit taught in the course being only about one week in duration, it could have been beneficial to spread it out over a longer duration to allow for further discussion. The results obtained from the study show that one week is long enough to increase individuals' awareness surrounding neurodiversity and their likelihood to disclose their neurodivergent status, but not enough time to shift their attitudes and beliefs surrounding ableism.

Increasing the duration of the neurodiversity module may provide further opportunity to shift students' attitudes and beliefs about ableism. An expanded module could afford opportunities for students to engage in acts of critical consciousness surrounding ableism. Critical consciousness consists of three main components 1) critical reflection, 2) political efficacy, and 3) critical action. Implementing engagement opportunities to practice these components may increase self-efficacy and awareness of how to identify ableism. An example engagement opportunity could include identifying an aspect of university policy that negatively impacts disabled and neurodivergent individuals (critical reflection), then create an updated version of this policy that provides more inclusion and awareness of disabled and neurodivergent individuals – increasing confidence to facilitate change (political efficacy), then sharing this updated policy with university administration (critical action).

Pedagogically, educators should consider integrating neurodiversity education across various courses and disciplines, rather than confining it to a single module within diversity courses. Additionally, training for educators on how to effectively teach about neurodiversity and ways to challenge ableism are crucial. With this, community-based participatory research could benefit the updated neurodiversity module. Including more neurodivergent voices in the recreation of this module would provide a deeper understanding of what neurodivergent individuals feel is important for fellow students to know about neurodiversity and ableism. This community knowledge could strengthen module components such as what materials (books, articles, podcasts) are presented to students to ensure the neurodivergent experience is accurately represented and that discussions around neurodiversity are comprehensive, inclusive, and transformative.

It is important to provide neurodivergent individuals with resources that they can utilize, which should be incorporated into the neurodiversity module. Although neurodiversity has often not been in the forefront of clinical practice and therapeutic discussions, it is important to understand and continue to invest in neurodivergence-informed therapy [34]. This type of therapy emphasizes epistemic humility (e.g., recognizing places where knowledge may be lacking), adopting a relational approach to disability (i.e., concerns are not inherent to the neurodivergent individual), diverging from normalization, and promoting pride and community in disabilities [34]. Neurodivergent individuals should also be encouraged and provided with ways to engage in clinical training as well [34].

As seen through study results, as students learned about neurodiversity, self-endorsed neurodivergence increased. However, the current module doesn't provide further information on how to access institutional or external supports – potentially leaving students in an "Okay, now what?" position. Future iterations of the neurodiversity module should include information on campus resources, how to receive a diagnosis (if so desired), and support for symptoms they may be experiencing. These additional resources could support the impact of the module even after its completion.

## Limitations

Following the work of Clarke, Schiavone, and Vazire [35], this limitations section is organized by validity type.

### External validity

The findings from this study may have limited external validity given the minimal race, age, sexual orientation, and gender diversity present in this sample. The utilized sample was predominantly White/European American, young adults, heterosexual, and female.

Some of the bias embedded in research can be mitigated with the use of participatory research – consulting and collaborating with individuals within communities of interest [36–38]. We acknowledge the importance of incorporating individuals within communities of interest; however, the current study did not explicitly use participatory research methods and/or gather input from neurodivergent individuals at each stage in the research process. This may limit study external validity to neurodivergent populations. Future studies can aim to mitigate this limitation by utilizing participatory research methods and/or by actively seeking feedback from neurodivergent individuals.

This study primarily focuses on more commonly assessed neurodivergent identities – limiting its external validity to the wider neurodivergent populous. Future work can improve on this by explicitly incorporating more underrepresented neurodivergent identities.

### Construct validity

The index of neurodiversity knowledge was limited to one item – although it had face validity – it did not capture the range of growth or the accuracy of student reports. Students were less likely to report 'I don't know' what neurodiversity means in the post survey, but this did not capture if their perceptions were accurate or the depth of this gained knowledge.

### Internal validity

The design of this study did not randomize students into classes with and without the neurodiversity module which would allow for more causal conclusions. Changes in student knowledge, self-endorsements, and critical consciousness of ableism cannot be causally linked to the neurodiversity module.

### Statistical conclusion validity

The limited variability in this sample, with respect to race, sexual orientation, and gender required us to dichotomize these identity variables. Although this dichotomization preserved power in the analyses it limited our ability to detect meaningful differences based on these population factors. The null results with respect to neurodivergent identity self-reports across race, sexual orientation, and gender may reflect our sample's limited variability.

### Future directions

The present study contributes to the growing body of literature on neurodiversity in HE by demonstrating potential benefits of incorporating neurodiversity-focused modules in undergraduate courses. While the module successfully increased awareness and self-disclosure of neurodivergent identities, the unexpected decrease in critical consciousness of ableism underscores the need for a more nuanced and critical approach to teaching about neurodiversity. By addressing these gaps, HE institutions can better support neurodivergent students and foster more inclusive academic environments.

## Acknowledgments

We would like to thank Monica Barany, Melanie Clayton, and the students of the Sleep and Developmental Studies Laboratory at Purdue University for their data cleaning and manuscript preparation support. We are also grateful to Amy Childress and Craig Zywicki from Purdue's Office of Undergraduate Research who supported the development of this CURE (course-based undergraduate research experience) and continue to support our data collection. This acknowledgment would not be complete without a sincere thank you to the students of Human Development and Family Science (HDFS) 280: Diversity in the Individual and the Family – who made this study possible.

## Author contributions

**Conceptualization:** A.J. Schwichtenberg, Katherine Mirah, Amy Janis, Moon Wests, Annabelle L. Atkin.

**Data curation:** A.J. Schwichtenberg, Annabelle L. Atkin.

**Formal analysis:** A.J. Schwichtenberg, Katherine Mirah, Amy Janis.

**Investigation:** A.J. Schwichtenberg, Annabelle L. Atkin.

**Methodology:** A.J. Schwichtenberg.

**Project administration:** A.J. Schwichtenberg, Amy Janis.

**Supervision:** A.J. Schwichtenberg, Amy Janis, Moon Wests.

**Writing – original draft:** A.J. Schwichtenberg, Katherine Mirah, Amy Janis.

**Writing – review & editing:** A.J. Schwichtenberg, Katherine Mirah, Amy Janis, Moon Wests, Annabelle L. Atkin.

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
