## [Decision Letter · Decision Letter 0]

PONE-D-24-52912Diversity in Education Study (DivES):  Explorations in Higher Education NeurodiversityPLOS ONE

Dear Dr. Schwichtenberg,

Thank you for submitting your manuscript to PLOS ONE. After careful consideration, we feel that it has merit but does not fully meet PLOS ONE’s publication criteria as it currently stands. Therefore, we invite you to submit a revised version of the manuscript that addresses the points raised during the review process.

I am sorry for the delay in securing reviews for this manuscript.I am very grateful to the 2 reviewers who have given considered, expert feedback. Please ensure that you highlight substantial changes in your revised manuscript, and also to provide a detailed response to reviewers' comments.==============================

We look forward to receiving your revised manuscript.

Kind regards,

Christopher James Hand, Ph.D., M.Sc., M.A., PgCAP

Academic Editor

PLOS ONE

Journal Requirements: 

**Comments from PLOS Editorial Office** : We note that one or more reviewers has recommended that you cite specific previously published works. As always, we recommend that you please review and evaluate the requested works to determine whether they are relevant and should be cited. It is not a requirement to cite these works and you may remove them before the manuscript proceeds to publication. We appreciate your attention to this request.

4. In the online submission form, you indicated that the data underlying the results presented in the study are available from Dr. A.J. Schwichtenberg. They can be released after publication.

Additional Editor Comments :

I am sincerely sorry for the delay in getting feedback to you.

I am very grateful to the two Reviewers who have provided their feedback and recommendations.

Please ensure that you highlight substantial changes within your revised manuscript, as well as providing a point-by-point response to reviewers.

I wish you well with your revisions.

Reviewers' comments:

Reviewer's Responses to Questions

**Comments to the Author**

1. Is the manuscript technically sound, and do the data support the conclusions?

Reviewer #1: Yes

Reviewer #2: Yes

2. Has the statistical analysis been performed appropriately and rigorously? 

Reviewer #1: Yes

Reviewer #2: I Don't Know

3. Have the authors made all data underlying the findings in their manuscript fully available?

Reviewer #1: No

Reviewer #2: No

4. Is the manuscript presented in an intelligible fashion and written in standard English?

Reviewer #1: Yes

Reviewer #2: Yes

5. Review Comments to the Author

Reviewer #1: This manuscript investigates the impact of a neurodiversity-focused module on college students. The study aimed to determine whether such a module could increase neurodiversity awareness, enhance self-endorsements, and shift attitudes toward ableism. The authors found that the module led to an increase in both neurodiversity awareness and self-endorsements. However, it also resulted in a negative shift in critical consciousness of ableism scores.

The basic motivation of the manuscript is commendable, as it identifies some gaps in the existing research. However, there are several areas of the manuscript that require further attention.

Introduction

(1) The introduction explains the concept of neurodiversity, listing various neurodivergences such as autism spectrum disorder, attention deficit hyperactivity disorder, dyslexia, and dyspraxia. To provide a comprehensive overview, please reference Monique Bottema-Beutel, Robert Chapman, and Elsherif et al. (2022). It's important to distinguish this broader concept of neurodiversity from more specific usages like the 'neurodiversity paradigm'.

Botha, M., Chapman, R., Giwa Onaiwu, M., Kapp, S. K., Stannard Ashley, A., & Walker, N. (2024). The neurodiversity concept was developed collectively: An overdue correction on the origins of neurodiversity theory. Autism, 28(6), 1591-1594. https://doi.org/10.1177/13623613241237871

Chapman, R. (2021). Neurodiversity and the social ecology of mental functions. Perspectives on Psychological Science, 16(6), 1360-1372.

Elsherif, M. M., Middleton, S. L., Phan, J. M., Azevedo, F., Iley, B., Grose-Hodge, M., ... & Dokovova, M. (2022). Bridging neurodiversity and open scholarship: How shared values can guide best practices for research integrity, social justice, and principled education. MetaArXiv. June, 20.

Pellicano, E. & denHouting, J. (2022) Annual research review: shifting from 'normal science' to neurodiversity in autism science. Journal of Child Psychology and Psychiatry, 63(4), 381-396.

(2) Several phrases in the manuscript indicate a lack of familiarity with the literature on neurodiversity. For example, the term 'neurodivergent' is used to refer to individuals who are different from neurotypical individuals. However, 'neurodiversity' is a broader concept that encompasses all neurological variations, including both neurotypical and neurodivergent individuals. The manuscript repeatedly conflates these terms, leading to potential inaccuracies- please see Shaw et al. (2024) and would recommend the papers for medical students which highlight the challenges they face in higher education as well (Shaw et al., 2018, 2023) and anxiety between groups (Mahak et al., 2024).

Godfrey-Harris, M., & Shaw, S. C. K. (2023). The experiences of medical students with ADHD: A phenomenological study. PLOS ONE, 18(8), e0290513–e0290513. https://doi.org/10.1371/journal.pone.0290513

Shaw, S. C. K., Brown, M. E. L., Jain, N. R., George, R. E., Bernard, S., Godfrey‐Harris, M., & Doherty, M. (2024). When I say … neurodiversity paradigm. Medical Education. https://doi.org/10.1111/medu.15565

Shaw, S. C. K., & Anderson, J. L. (2018). The experiences of medical students with dyslexia: An interpretive phenomenological study. Dyslexia, 24(3), 220–233. https://doi.org/10.1002/dys.1587

Mahak, S., Malone, S. A., Elsherif, M. M., Hand, C. J., & Morsanyi, K. (2024, November 21). Do neurodivergent individuals have higher statistics and mathematics anxiety? Evaluating evidence from a large, multi-lab study. Retrieved from osf.io/preprints/psyarxiv/zhx8k

(3). The manuscript acknowledges the neurodiversity paradigm, which views differences in cognition as natural biological variations. However, when referring to autism spectrum disorder and attention deficit hyperactivity disorder (ADHD), it is more appropriate to use the term "autism" to align with community preference. Additionally, it is important to recognize that individuals with ADHD may prefer alternative terms, such as "attention dysregulation hyperactivity disorder" (Dwyer et al., 2024).

Dwyer, P., Williams, Z. J., Lawson, W., & Rivera, S. M. (2024). A trans-diagnostic investigation of attention, hyper-focus, and monotropism in autism, ADHD, and the general population. Neurodiversity. Advance online publication. https://doi.org/10.1177/27546330241237883

(4) In the section on ableism, a definition of ableism would help make this easier for people to interpret, as people often make assumptions about what ableism entails. Constantino, Campbell, and Simpson (2022) provide an excellent definition of ableism and raise important points about the social model and how to conceptualize ableism within this framework.

Constantino, C., Campbell, P., & Simpson, S. (2022). Stuttering and the social model. Journal of Communication Disorders, 96, 106200.

(5) It was unclear whether this study was confirmatory or exploratory. If confirmatory, please state the specific hypotheses being tested. If exploratory, please clearly indicate that the study is exploratory in nature.

Methods

Found the section about IRB exemption category II quite easy to follow and an important point to highlight. I would recommend clarifying the first sentence, as it could be misinterpreted negatively. The subsequent statement within the same paragraph seems to contradict this initial point.

"This study meets IRB exemption category two; wherein, informed consent procedures are not required given all data were collected via an educational survey."

"However, "going beyond minimum requirements, the first question within the survey asks for student informed written consent – which includes study purpose, length, risks, benefits, and confidentiality. Opportunities for consent-related questions were provided in-person during class. Instructors were blind to student consent status (i.e., They knew all students completed the survey for class but were not aware of who consented for their data to be used within this study)"

You can write it as:

This study was exempted from full IRB review under category two, which allows for the waiver of informed consent procedures when data is collected through educational surveys. However, to ensure transparency and respect for participant autonomy, the initial survey question sought explicit written informed consent from students. This consent form outlined the study's purpose, duration, potential risks and benefits, and confidentiality measures. To maintain the integrity of the study, instructors were unaware of individual student consent status, ensuring that all students completed the survey as part of their coursework.

Please provide Cronbach's Alpha and McDonald's Omega coefficients for the Contemporary Critical Consciousness Measure II (CCCMII; Shin et al., 2018) to assess its reliability.

The inclusion of a positionality statement is becoming increasingly common practice in research, particularly in fields that emphasize reflexivity. To enhance transparency regarding the researcher's standpoint and potential biases, consider including a brief positionality statement in the manuscript. This could disclose the authors' relevant experiences or affiliations that might influence the interpretation of the data.

While this study did not explicitly involve participatory research methods or seek input from neurodivergent individuals to verify findings, it is important to acknowledge whether this was done or not. Participatory research can empower research participants, foster collaboration, and enhance the validity and relevance of research outcomes. Additionally, involving neurodivergent individuals in the interpretation of findings can provide valuable perspectives and help to mitigate potential biases.

Here are some references to cite:

Bourke, L. (2009). Reflections on doing participatory research in health: Participation,

method and power. International Journal of Social Research Methodology, 12(5), 457-

474. https://doi.org/10.1080/13645570802373676

Fletcher-Watson, S., Brook, K., Hallett, S., Murray, F., & Crompton, C. J. (2021). Inclusive

practices for neurodevelopmental research. Current Developmental Disorders Reports,

8(2), 88-97. https://doi.org/10.1007/s40474-021-00227-z

Gourdon-Kanhukamwe, A., Kalandadze, T., Yeung, S. K., Azevedo, F., Iley, B., Phan, J. M., ... & Elsherif, M. M. (2023). Opening up understanding of neurodiversity: a call for applying participatory and open scholarship practices. The Cognitive Psychology Bulletin, 8.

Locate the paper here: https://research.edgehill.ac.uk/ws/portalfiles/portal/71535551/Gourdon_Kanhukamwe_etal_CPB_2023_Opening_up_understanding_of_neurodiversity.pdf

(5) Results

Overall, it would be beneficial to provide comprehensive details regarding the range, skewness, kurtosis, and normality of the distribution. This information could be included within the main text or supplementary materials.

While the focus on p-values is understandable, it's important to remember that statistical significance alone does not necessarily equate to practical significance. To demonstrate the real-world impact of the findings, considering effect sizes, as suggested by Skottun and Skoyles (2012), would be valuable.

Skottun, B. C., & Skoyles, J. R. (2012). Interletter spacing and dyslexia. Proceedings of the National Academy of Sciences, 109(44), E2958-E2958.

Please expand upon the following points to provide a more detailed and nuanced understanding of the broader context by including statistical tests, degrees of freedom and effect sizes.

Certain claims lack much information that makes it hard to understand what the differences were. For instance, In the case of the CCMII ableism scales, differences by instructor emerged. Differences by instructor were also investigated for the other research questions, but it did not influence the other results.

To ensure robust analysis and mitigate potential bias, it is essential to report on missingness patterns and conduct tests to assess whether data are missing completely at random (MCAR), missing at random (MAR), or not missing at random (MNAR) (Woods et al., 2023).

Woods, A. D. et al.(2023). Best practices for addressing missing data through multiple imputation. Infant and Child Development, e2407.

Little, R. J. A. (1988) A Test of Missing Completely at Random for Multivariate Data with Missing Values. J. Am. Stat. Assoc. 83, 1198–1202.

Discussion

Please include one more neurodivergence here to make it clear that neurodiversity is not limited to these specific groups, as it tends to be seen by some researchers who investigated neurodiversity and others that neurodiversity include only autism and ADHD, it would be helpful to dismantle this stereotype:

However, in Purdue’s Student Experience Survey roughly 36% of the incoming student cohort endorsed a category that could fit under the neurodiversity umbrella (e.g., autism, ADHD).

Focus on the most significant limitations, contextualizing them specifically. Explain how these limitations may introduce bias, uncertainty, or alternative explanations. The more specific you can be about the implications, the better. Put simply, provide concrete explanations (read Clarke et al., 2024 Social and Personality Psychology Compass).

https://compass.onlinelibrary.wiley.com/doi/full/10.1111/spc3.12979

It would also be beneficial to include some points on the limitations of current research, highlighting the need to focus on neurodivergences that have received less attention, such as Tourette's Syndrome, Dyscalculia, Developmental Language Disorder (DLD), stammering, dyspraxia, and others (see Figure 1 in Layinka et al., 2024).

Layinka, O., Hargitai, L. D., Shah, P., Waldren, L. H., & Leung, F. Y. (2024). Five interdisciplinary tensions and opportunities in neurodiversity research. Elife, 13, e98461.

Additionally, a positive outlook should be considered. A recommended reference by Chapman and Botha (2022) exploring Neurodivergence-informed therapy could be a valuable addition. Their work emphasizes the therapist's critical role in driving systemic societal change and reframing deficits as relational rather than individual issues. The aim is to empower neurodivergent individuals by challenging neuro-normative expectations and cultivating relational epistemic humility, which involves acknowledging the diverse experiences of neurodivergence and disability. By focusing on quality of life, this approach can contribute to a more inclusive and equitable educational landscape.

Chapman, R., & Botha, M. (2023). Neurodivergence‐informed therapy. Developmental Medicine & Child Neurology, 65(3), 310-317. https://doi.org/10.1111/dmcn.15384

Reviewer #2: Thanks for the opportunity to review the article Diversity in Education Study (DivES): Explorations in Higher Education Neurodiversity. This paper examined the effect of a week-long neurodiversity-focused module on student understanding of the term neurodiversity, self-endorsements of neurodiversity, and critical consciousness of ableism.

I agree with the authors that it is an important topic, and has been less examined in higher educational settings. This study addresses this specific research gap.

Overall, the paper is well written and easy to follow through.

The introduction has provided what we currently know in this area, explained and defined the constructs (e.g., ableism) well. The method has explained the design well, and with a good description of the teaching module. In the discussion section, the authors also provided explanation why some of the unexpected results were presented and its implication for future pedagogical design.

Here are a few minor comments for the authors to consider:

Introduction

When the authors have addressed the key concepts well, and why it is important to reduce ableism in HE. The current study focuses on using a specific teaching module to achieve these goals. It would be great that if the authors could further explain the rationale of why and how an (this particular) education module may bridge this gap. Currently, these concepts (a weeklong module of neurodiversity shift reports of critical conscious of ableism) have not been clearly explored and explained.

Method

P.14

'Data for three students were excluded from analyses because they did not consistently endorse their

professor’s name in the pre-post surveys or they endorsed a professor that was not an option at

that point in time.'

Can the authors please clarify what it means 'endorse' a professor name and why it leads to an exclusion because of these? At the moment, it is a bit unclear to the readers, especially, when it is listed early in the method section.

P.18 can the authors explain what is the rationale of using 'heterosexual vs other sexual orientation endorsements as covariates?

Results

Because the structure of the current paper does not have a data analysis section, so in the results section, suggest the authors can clearly mention the type of stat analyses were used.

e.g. p. 18, …

Neurodiversity endorsement did not systematically vary by gender in the pre-survey, X2 (1) = 1.80, p = .18 or the post-survey, X2 (1) = .001, p = .99…

A stat method was only clearly mentioned in p.19 'A paired-sample t-test was used to assess change from the start to the end of the term.', but was not clearly mentioned across other part of the results section.

When there is a statement at the end of method section mentioned, 'Each of these were coded as most common features vs. less common feature (e.g., heterosexual vs. all other sexual orientation endorsements) and were considered as covariates in the analyses.' It is unclear that how these variables were used as covariates in the results section.

Discussion section

Whereas the authors provide practical implication (Pedagogical reflections) of the current paper, it would be good to see if the authors could discuss some 'future research direction'. It seems to be important, as the authors mentioned that it's an important areas and less researched; and the authors also highlighted some limitations of the current studies. Suggest future research direction could be reflected on the current findings and the limitation of the current methodologies.

I am aware that the authors mentioned, 'Replication of this study with larger and more diverse samples is necessary to validate these preliminary findings and refine the neurodiversity module', would there be any other ways to improve the current research design so a more conclusive statement could be made?

Thank you!

6. PLOS authors have the option to publish the peer review history of their article (what does this mean? ). If published, this will include your full peer review and any attached files.

**Do you want your identity to be public for this peer review?** For information about this choice, including consent withdrawal, please see our Privacy Policy .

Reviewer #1: No

Reviewer #2: No

---

## [Author Response · Author response to Decision Letter 1]

2 May 2025

We thank the reviewers for taking the time to provide detailed and supportive feedback. We have, for the most part, followed all the provided recommendations and we feel the manuscript is now far improved – thank you.

Editorial

Editorial comment: PLOS ONE Style Requirements

Response: The manuscript is now adjusted to fit PLOS ONE style requirements.

Editorial comment: PLOS ONE data share compliance

Response: All data associated with this manuscript will be publicly available at https://osf.io/p98dc/ (doi:10.17605/OSF.IO/P98DC) upon publication.

Reviewer 1, Comment 1: This manuscript investigates the impact of a neurodiversity-focused module on college students. The study aimed to determine whether such a module could increase neurodiversity awareness, enhance self-endorsements, and shift attitudes toward ableism. The authors found that the module led to an increase in both neurodiversity awareness and self-endorsements. However, it also resulted in a negative shift in critical consciousness of ableism scores.

The basic motivation of the manuscript is commendable, as it identifies some gaps in the existing research.

Response Reviewer 1, Comment 1: Thank you for recognizing and noting some of our manuscript/study strengths.

Reviewer 1, Comment 2: The introduction explains the concept of neurodiversity, listing various neurodivergences such as autism spectrum disorder, attention deficit hyperactivity disorder, dyslexia, and dyspraxia. To provide a comprehensive overview, please reference Monique Bottema-Beutel, Robert Chapman, and Elsherif et al. (2022). It's important to distinguish this broader concept of neurodiversity from more specific usages like the 'neurodiversity paradigm'. [References provided: Botha et al., 2024; Chapman, 2021; Elsherif et al., 2022; Pellicano & denHouting., 2022)

Response Reviewer 1, Comment 2: We thank this reviewer for this constructive comment and the provided sources. We now incorporate all of the provided references and the introduction was adjusted to include a clearer picture of neurodiversity and the neurodiversity paradigm. Additionally, the introduction is updated to more accurately reflect the role of both neurodivergent and neurotypical individuals in increasing knowledge of the neurodiversity paradigm and ableism in higher education.

Reviewer 1, Comment 3: Several phrases in the manuscript indicate a lack of familiarity with the literature on neurodiversity. For example, the term 'neurodivergent' is used to refer to individuals who are different from neurotypical individuals. However, 'neurodiversity' is a broader concept that encompasses all neurological variations, including both neurotypical and neurodivergent individuals. The manuscript repeatedly conflates these terms, leading to potential inaccuracies- please see Shaw et al. (2024) and would recommend the papers for medical students which highlight the challenges they face in higher education as well (Shaw et al., 2018, 2023) and anxiety between groups (Mahak et al., 2024). [References provided: Godfrey-Harris & Shaw, 2023; Shaw et al., 2024; Shaw & Anderson, 2018; Mahak et al., 2024]

Response Reviewer 1, Comment 3: We fully agree with this comment. The paper has been reviewed to ensure proper usage of “neurodiverse” and “neurodivergent” throughout. Due to the nature of specific survey items, there are places where neurodiverse is still used in the manuscript where neurodivergence may be more accurate. Additionally, articles discussing the experiences of ADHD and dyslexic medical students have been added to the introduction section to provide further context on the role of educational systems in association with neurodivergent student experiences.

Reviewer 1, Comment 4: The manuscript acknowledges the neurodiversity paradigm, which views differences in cognition as natural biological variations. However, when referring to autism spectrum disorder and attention deficit hyperactivity disorder (ADHD), it is more appropriate to use the term "autism" to align with community preference. Additionally, it is important to recognize that individuals with ADHD may prefer alternative terms, such as "attention dysregulation hyperactivity disorder" (Dwyer et al., 2024).

Response Reviewer 1, Comment 4: The paper has been updated to include attention dysregulation hyperactivity disorder along with ADHD and throughout the manuscript autism is now used to describe autistic individuals.

Reviewer 1, Comment 5: In the section on ableism, a definition of ableism would help make this easier for people to interpret, as people often make assumptions about what ableism entails. Constantino, Campbell, and Simpson (2022) provide an excellent definition of ableism and raise important points about the social model and how to conceptualize ableism within this framework.

Response Reviewer 1, Comment 5: Thank you for this suggestion and the resource. We now include a definition of ableism that builds on Constantino et al (2022) in the introduction.

Reviewer 1, Comment 6: It was unclear whether this study was confirmatory or exploratory. If confirmatory, please state the specific hypotheses being tested. If exploratory, please clearly indicate that the study is exploratory in nature.

Response Reviewer 1, Comment 6: Thank you for this comment. We now explicitly outline our research questions and hypothesis in the current student section. We’ve also removed explorations from the title.

Reviewer 1, Comment 7: Methods: Found the section about IRB exemption category II quite easy to follow and an important point to highlight. I would recommend clarifying the first sentence, as it could be misinterpreted negatively. The subsequent statement within the same paragraph seems to contradict this initial point.

"This study meets IRB exemption category two; wherein, informed consent procedures are not required given all data were collected via an educational survey."

"However, "going beyond minimum requirements, the first question within the survey asks for student informed written consent – which includes study purpose, length, risks, benefits, and confidentiality. Opportunities for consent-related questions were provided in-person during class. Instructors were blind to student consent status (i.e., They knew all students completed the survey for class but were not aware of who consented for their data to be used within this study)"

You can write it as:

This study was exempted from full IRB review under category two, which allows for the waiver of informed consent procedures when data is collected through educational surveys. However, to ensure transparency and respect for participant autonomy, the initial survey question sought explicit written informed consent from students. This consent form outlined the study's purpose, duration, potential risks and benefits, and confidentiality measures. To maintain the integrity of the study, instructors were unaware of individual student consent status, ensuring that all students completed the survey as part of their coursework.

Response Reviewer 1, Comment 7: Thank you for this constructive comment and text recommendations. The manuscript is now updated as recommended.

Reviewer 1, Comment 8: Please provide Cronbach's Alpha and McDonald's Omega coefficients for the Contemporary Critical Consciousness Measure II (CCCMII; Shin et al., 2018) to assess its reliability.

Response Reviewer 1, Comment 8: Thank you for the recommendation, these reliability estimates are now added to the revised manuscript.

Reviewer 1, Comment 9: The inclusion of a positionality statement is becoming increasingly common practice in research, particularly in fields that emphasize reflexivity. To enhance transparency regarding the researcher's standpoint and potential biases, consider including a brief positionality statement in the manuscript. This could disclose the authors' relevant experiences or affiliations that might influence the interpretation of the data.

Response Reviewer 1, Comment 9: Thank you for this comment. We agree that it is important to disclose positionality to ensure transparency for readers. We now include a positionality statement.

Reviewer 1, Comment 10: While this study did not explicitly involve participatory research methods or seek input from neurodivergent individuals to verify findings, it is important to acknowledge whether this was done or not. Participatory research can empower research participants, foster collaboration, and enhance the validity and relevance of research outcomes. Additionally, involving neurodivergent individuals in the interpretation of findings can provide valuable perspectives and help to mitigate potential biases. Here are some references to cite: [Bourke, 2009; Fletcher-Watson et al., 2021; Gourdon-Kanhukamwe et al., 2023]

Response Reviewer 1, Comment 10: Thank you for this thoughtful comment and the sources provided. Our team includes members with neurodivergent identities, and we have incorporated their feedback on finding interpretation and have added this to our positionality statement.

Reviewer 1, Comment 11: Overall, it would be beneficial to provide comprehensive details regarding the range, skewness, kurtosis, and normality of the distribution. This information could be included within the main text or supplementary materials.

Response Reviewer 1, Comment 11: Most of the metrics utilized within this study were frequency based or categorical (e.g., gender, race, neurodivergent identity). We’ve provided count and percentage summaries in Tables 1, 2, and 3. Additionally, to be responsive to this comment, we’ve added the mean, standard deviation, range, skewness, kurtosis and confirmed a normal distribution in our continuous measure of ableism.

Reviewer 1, Comment 12: While the focus on p-values is understandable, it's important to remember that statistical significance alone does not necessarily equate to practical significance. To demonstrate the real-world impact of the findings, considering effect sizes…

Response Reviewer 1, Comment 12: We fully agree, we’ve added Phi (φ) for all X2 tests and Cohen’s d for the paired t-tests.

Reviewer 1, Comment 13: Please expand upon the following points to provide a more detailed and nuanced understanding of the broader context by including statistical tests, degrees of freedom and effect sizes.

Certain claims lack much information that makes it hard to understand what the differences were. For instance, In the case of the CCMII ableism scales, differences by instructor emerged. Differences by instructor were also investigated for the other research questions, but it did not influence the other results.

Response Reviewer 1, Comment 13: To address this comment we’ve added an analytic plan section, effect sizes, and we’ve expanded and moved our post-hoc professor analyses to the results section.

Reviewer 1, Comment 14: To ensure robust analysis and mitigate potential bias, it is essential to report on missingness patterns and conduct tests to assess whether data are missing completely at random (MCAR), missing at random (MAR), or not missing at random (MNAR) (Woods et al., 2023).

Response Reviewer 1, Comment 14: We had minimal missingness within this study. A minimum of two missingness patterns are needed to assess MCAR – with only 2% missing data points across all variables – we did not meet this minimum requirement.

Reviewer 1, Comment 15: Please include one more neurodivergence here to make it clear that neurodiversity is not limited to these specific groups, as it tends to be seen by some researchers who investigated neurodiversity and others that neurodiversity include only autism and ADHD, it would be helpful to dismantle this stereotype:

However, in Purdue’s Student Experience Survey roughly 36% of the incoming student cohort endorsed a category that could fit under the neurodiversity umbrella (e.g., autism, ADHD).

Response Reviewer 1, Comment 15: We absolutely agree. Neurodiversity encompasses many identities. With this, we have added more neurodivergent identities to this section.

Reviewer 1, Comment 16: Focus on the most significant limitations, contextualizing them specifically. Explain how these limitations may introduce bias, uncertainty, or alternative explanations. The more specific you can be about the implications, the better. Put simply, provide concrete explanations (read Clarke et al., 2024 Social and Personality Psychology Compass).

Response Reviewer 1, Comment 16: An excellent paper – thank you for the recommendation. We’ve reorganized our limitations section building on the structure provided in this resource.

Reviewer 1, Comment 17: It would also be beneficial to include some points on the limitations of current research, highlighting the need to focus on neurodivergences that have received less attention, such as Tourette's Syndrome, Dyscalculia, Developmental Language Disorder (DLD), stammering, dyspraxia, and others (see Figure 1 in Layinka et al., 2024).

Response Reviewer 1, Comment 17: Thank you for highlighting the need for increased inclusivity and awareness in neurodiversity research. We’ve added to the limitations section addressing how this study primarily focuses on more “commonly” assessed neurodivergent identities. A call for future work to include underrepresented neurodivergent identities in research is now also included.

Reviewer 1, Comment 18: Additionally, a positive outlook should be considered. A recommended reference by Chapman and Botha (2022) exploring Neurodivergence-informed therapy could be a valuable addition. Their work emphasizes the therapist's critical role in driving systemic societal change and reframing deficits as relational rather than individual issues. The aim is to empower neurodivergent individuals by challenging neuro-normative expectations and cultivating relational epistemic humility, which involves acknowledging the diverse experiences of neurodivergence and disability. By focusing on quality of life, this approach can contribute to a more inclusive and equitable educational landscape.

Response Reviewer 1, Comment 18: We appreciate this comment as it is important to instill hope and optimism. We have incorporated the reference provided here into the Pedagogical Reflections section of the manuscript.

Reviewer 2:

Reviewer 2, Comment 1: I agree with the authors that it is an important topic, and has been less examined in higher educational settings. This study addresses this specific research gap.

Overall, the paper is well written and easy to follow through.

The introduction has provided what we currently know in this area, explained and defined the constructs (e.g., ableism) well. The method has explained the design well, and with a good description of the teaching module. In the discussion section, the authors also provided explanation why some of the unexpected results were presented and its implication for future pedagogical design.

Response Reviewer 2, Comment 1: Thank you for considering and noting some of our manuscript strengths.

Reviewer 2, Comment 2:

When the authors have addressed the key concepts well, and why it is important to reduce ableism in HE. The current study focuses on using a specific teaching module to achieve these goals. It would be great that if the authors could further explain the rationale of why and how an (this particular) education module may bridge this gap. Currently, these concepts (a weeklong module of neurodiversity shift reports of critical conscious of ableism) have not been clearly explored and explained.

Response Reviewer 2, Comment 2: Thank you for this comment. In hindsight we were not surprised our neurodiversity module did not ‘move the needle’ on the recognition of ableism in HE. Our inclusion of the Contemporary Critical Consciousness Measure II is a reflection of the larger project - Diversity in Education Study (DivES). Within DivES several systems of oppression, discrimination, and prejudice are presented with respect to race, ethnicity, sexual orientation, and several additional identities. Although systems of ableism were not explicitly taught it seemed reasonable, at the time of study design, that the consciousness of ableism could increase with the conscio

---

## [Decision Letter · Decision Letter 1]

Diversity in Education Study (DivES):

Investigating a Neurodiversity Module in Higher Education

PONE-D-24-52912R1

Dear Dr. Schwichtenberg,

We’re pleased to inform you that your manuscript has been judged scientifically suitable for publication and will be formally accepted for publication once it meets all outstanding technical requirements.

Kind regards,

Maheshkumar Baladaniya

Academic Editor

PLOS ONE

Additional Editor Comments (optional):

Authors have made changes in the revision which make manuscript acceptable for the publication.

Reviewers' comments:

Reviewer's Responses to Questions

**Comments to the Author**

1. If the authors have adequately addressed your comments raised in a previous round of review and you feel that this manuscript is now acceptable for publication, you may indicate that here to bypass the “Comments to the Author” section, enter your conflict of interest statement in the “Confidential to Editor” section, and submit your "Accept" recommendation.

Reviewer #1: All comments have been addressed

Reviewer #2: All comments have been addressed

Reviewer #3: All comments have been addressed

2. Is the manuscript technically sound, and do the data support the conclusions?

Reviewer #1: Yes

Reviewer #2: Yes

Reviewer #3: Yes

3. Has the statistical analysis been performed appropriately and rigorously? 

Reviewer #1: Yes

Reviewer #2: Yes

Reviewer #3: Yes

4. Have the authors made all data underlying the findings in their manuscript fully available?

Reviewer #1: Yes

Reviewer #2: Yes

Reviewer #3: Yes

5. Is the manuscript presented in an intelligible fashion and written in standard English?

Reviewer #1: Yes

Reviewer #2: Yes

Reviewer #3: Yes

6. Review Comments to the Author

Reviewer #1: The authors have made significant improvements to the manuscript following the previous review. The present study investigated the impact of a neurodiversity focused module on college students. The study aimed to determine whether such a module could increase neurodiversity awareness, enhance self-endorsements, and shift attitudes toward ableism. The authors found that the module led to an increase in both neurodiversity awareness and self-endorsements. The revisions have addressed many of the concerns regarding clarity of rationale and the presentation of methods and results. The introduction now provides a more robust discussion of the relevant literature, and the method and results sections are considerably clearer. We commend the authors for doing such an amazing and excellent job. We thank them for taking these comments seriously.

Reviewer #2: Thanks for opportunity to re-review the manuscript, and thanks for the authors to consider all the comments carefully and made adjustment. As a result, it has substantially raised the standard of the manuscript.

In specific:

The introduction paragraph, hypotheses, clearly explained the rationale about that particular course may improve critical consciousness of ableism.

The analytic plan has clearly illustrated the statistical method used, which enhance readability.

While the authors have addressed all the comments I raised in the first revision satisfactorily, here is a major comment that I suggest the authors to consider (about the new text that was not presented in the first submission).

In the limitations section, I appreciate the author has taken the comment 16 from Reviewer 1 to discuss the limitations in more specific. I agree with this approach. The authors indeed did it relatively well in discussing External Validity section, by providing specific future research direction. However, for the sections Construct validity, Internal Validity, Statistical conclusion validity, the authors have only mentioned the limitations. Currently, the tone/approach of expression may make the readers to question the validity of the measures and hence the accuracy of the findings, and the integrity of the whole research. It might have prompted the readers to think the whole study was built on scales with poor internal validity, construct validity, and statistical conclusion validity.

In my opinion, a good limitation section should acknowledge the current limitation, and explain it further (here are some component for consideration, but not limited to): (1) why it really matters, and the results have to be interpreted with caution, (2) justify even it's a limitation, but it's not a big issue and did not affect the integrity of the findings, and how these risks have been mitigated, (3) illustrate the specific limitation and suggestion the need of future direction.

My suggestion is either: (1) the authors could discuss the limitation that are significant ONLY (which was also the wording Reviewer 1 used); (2) if the authors want to discuss the limitations of construct, internal, and statistical conclusion validity, it would be good to see the authors to justify whether those limitations are minor/major, whether there are still some strengths in the scale to mitigate the risk, and justify whether the authors consider the findings are meaningful and valid.

Another minor comment, instead of using the title 'Future directions', the authors may consider using the title 'Conclusion', or to add a 'Conclusion' section. Currently, readers may expect there is a conclusion section after future directions, and it's a bit rare to conclude a paper with a 'Future directions' section title only.

Reviewer #3: INTRODUCTION : Describe the potential theoretical impact of a neurodiversity module on attitudes connected to ableism; the connection is still somewhat unexplored.

POSITIONALITY STATEMENT : Elaborating on whether their input affected particular choices made regarding data analysis or instructional design.

METHODS : Go beyond a single item in the neurodiversity knowledge area; in subsequent iterations, think about including a brief multi-item knowledge scale. explaining why there was no explicit neurodiversity-specific material in the modified CCCMII to date. Justify the choice to use the pre-survey data to infer the outlier score.

RESULTS : Explain the open-ended response coding (were several raters involved? Was it possible to determine inter-rater agreements?

DISCUSSION : Explain the reduction in ableism scores in one group more clearly using a theoretical framework (e.g., potential response, short exposure). Make it clearer in the opening lines of this section that the shift in ableism awareness was unanticipated.

PEDAGOGICAL REFLECTIONS : Describe in further detail how including neurodiversity information over time could avoid these unexpected consequences. If any student comments or reflections were gathered, think about including them.

FUTURE DIRECTIONS : Encourage future modules to be co-designed with students who are neurodivergent. To determine whether shifts in self-identification are maintained, incorporate the creation of longitudinal tracking.

7. PLOS authors have the option to publish the peer review history of their article (what does this mean? ). If published, this will include your full peer review and any attached files.

**Do you want your identity to be public for this peer review?** For information about this choice, including consent withdrawal, please see our Privacy Policy .

Reviewer #1: No

Reviewer #2: No

Reviewer #3: No

---

## [Editor Report · Acceptance letter]

PONE-D-24-52912R1

PLOS ONE

Dear Dr. Schwichtenberg,

I'm pleased to inform you that your manuscript has been deemed suitable for publication in PLOS ONE. Congratulations! Your manuscript is now being handed over to our production team.

Kind regards,

on behalf of

Dr. Maheshkumar Baladaniya

Academic Editor

PLOS ONE